# Small Bowel Detection for Wireless Capsule Endoscopy Using Convolutional Neural Networks with Temporal Filtering

**DOI:** 10.3390/diagnostics12081858

**Published:** 2022-07-31

**Authors:** Geonhui Son, Taejoon Eo, Jiwoong An, Dong Jun Oh, Yejee Shin, Hyenogseop Rha, You Jin Kim, Yun Jeong Lim, Dosik Hwang

**Affiliations:** 1School of Electrical and Electronic Engineering, Yonsei University, Seoul 03722, Korea; higun2@yonsei.ac.kr (G.S.); ship9136@naver.com (T.E.); nanapow3@naver.com (J.A.); yejeeshin.ee@gmail.com (Y.S.); rhs970720@naver.com (H.R.); 2Department of Internal Medicine, Dongguk University Ilsan Hospital, Dongguk University College of Medicine, Goyang 10326, Korea; mileo31@naver.com; 3IntroMedic, Capsule Endoscopy Medical Device Manufacturer, Seoul 08375, Korea; ykim@intromedic.com; 4Center for Healthcare Robotics, Korea Institute of Science and Technology, 5, Hwarang-ro 14-gil, Seongbuk-gu, Seoul 02792, Korea; 5Department of Oral and Maxillofacial Radiology, Yonsei University College of Dentistry, Seoul 03722, Korea; 6Department of Radiology and Center for Clinical Imaging Data Science (CCIDS), Yonsei University College of Medicine, Seoul 03722, Korea

**Keywords:** capsule endoscopy, small bowel detection, convolutional neural networks, temporal filtering

## Abstract

By automatically classifying the stomach, small bowel, and colon, the reading time of the wireless capsule endoscopy (WCE) can be reduced. In addition, it is an essential first preprocessing step to localize the small bowel in order to apply automated small bowel lesion detection algorithms based on deep learning. The purpose of the study was to develop an automated small bowel detection method from long untrimmed videos captured from WCE. Through this, the stomach and colon can also be distinguished. The proposed method is based on a convolutional neural network (CNN) with a temporal filtering on the predicted probabilities from the CNN. For CNN, we use a ResNet50 model to classify three organs including stomach, small bowel, and colon. The hybrid temporal filter consisting of a Savitzky–Golay filter and a median filter is applied to the temporal probabilities for the “small bowel” class. After filtering, the small bowel and the other two organs are differentiated with thresholding. The study was conducted on dataset of 200 patients (100 normal and 100 abnormal WCE cases), which was divided into a training set of 140 cases, a validation set of 20 cases, and a test set of 40 cases. For the test set of 40 patients (20 normal and 20 abnormal WCE cases), the proposed method showed accuracy of 99.8% in binary classification for the small bowel. Transition time errors for gastrointestinal tracts were only 38.8 ± 25.8 seconds for the transition between stomach and small bowel and 32.0 ± 19.1 seconds for the transition between small bowel and colon, compared to the ground truth organ transition points marked by two experienced gastroenterologists.

## 1. Introduction

Since wireless capsule endoscopy (WCE) was first introduced in 2000 [1], small bowel capsule endoscopy (SBCE) has become a major modality for the diagnosis of various small bowel diseases because of its painless and non-invasive nature [2,3,4,5]. However, reading numerous image frames is a time-consuming and tedious task for clinicians [6]. In addition, there may be variations among readers and fatigue may also affect diagnostic accuracy [7].

Under these circumstances, artificial intelligence (AI) or deep learning algorithm [8,9] is becoming a technology that opens a new horizon in the reading of WCE. As AI algorithms have been used for lesion detection in various medical imaging modalities, papers on lesion detection in SBCE are also being actively published [10,11,12]. Various studies such as classifying organs, turbid, bubbles, clear blob, wrinkle, and wall and detecting bleeding or polyps have been presented [13,14,15,16,17]. They have been proven to be effective in diagnostic fields. Moreover, due to a long reading time of SBCE, clinicians expect the automated lesion detection algorithms to be a complementary tool that can shorten the reading time. 

For the AI-based automated reading of SBCE, it is important that the small bowel is preceded by automated classification as a preprocessing step. Furthermore, even if the AI algorithms are not directly developed for automated reading, it is possible to reduce the reading time for gastric transition time out of the total reading time by automated small bowel detection. Because (i) the capsule endoscope moves depending on gravity and gut peristalsis, and (ii) the pylorus and ileocecal valve, which are anatomic landmarks, are sphincter-like structures, it is rare for the capsule to regurgitate. Understanding these characteristics of SBCE, automatic anatomical distinction can be performed only for the small bowel by detecting the two transition points for stomach to small bowel and small bowel to colon. The most related work using this step is [17], which classifies single frame images into three organs including stomach, small bowel, and colon. However, this method cannot yield the organ transition points between the three organs.

Consequently, in this study, we proposed the automated organ transition point detection algorithm using a convolutional neural network (CNN) with temporal filtering. First, the CNN classifies each image taken from gastrointestinal tracts into the three classes including stomach, small bowel, and colon. Second, the temporal filter remarkably reduces the number of misclassified frames from the CNN by correcting each class probability of a frame using the long-term probabilities yielded from the adjacent frames. Finally, by thresholding the temporally filtered probabilities for the “small bowel” class, the frames from the small bowel, which are mapped to value 1, and the frames from the other two organs, which are mapped to value 0, are differentiated. Since organs were filmed in order, if the small bowel can be accurately classified, the stomach and colon are also classified. In this way, the proposed algorithm detects the two transition points for stomach to small bowel and small bowel to colon with long-term dependencies in WCE frames.

## 2. Materials and Methods

### 2.1. Study Design

#### 2.1.1. Data Acquisition

Two hundred and sixty WCE (MiroCam MC1000W and MC1200, Intromedic Co., Ltd., Seoul, Korea) cases performed at Dongguk University Ilsan Hospital between 2002 and 2022 were retrospectively acquired to be used for training and validation of the proposed small bowel detection method. All WCE images were extracted in JPEG format with matrix size of 320 × 320 and 3 fps using MiroView 4.0 (Intromedic Co., Ltd., Seoul, Korea). Cases with adult patients (>20 years) were included in this study. Cases in which the capsule endoscope did not pass into the colon and cases in which organs could not be visually distinguished were excluded. Our study was conducted with the approval of Institutional Review Board of Dongguk University Ilsan Hospital (DUIH 2022-01-032-001).

#### 2.1.2. Data Preparation

Two gastroenterologists specializing in capsule endoscopy (Oh DJ and Lim YJ from Dongguk University Ilsan Hospital) independently performed image labeling for organ classification. They read the whole images from each WCE case manually and marked them as stomach, small bowel, and colon, and cross-checked with each other. The 200 labeled case datasets were then classified into training (140 cases, 70%), validation (20 cases, 10%), and test (40 cases, 20%) set. The training set consisted of 1,572,274, 8,721,126, and 7,809,614 frames for stomach, small intestine, and colon, respectively. The validation set consisted of 136,266, 971,231, and 883,745 frames for stomach, small intestine, and colon, respectively. The test set consisted of 278,455, 2,046,571, and 1,541,448 frames for stomach, small intestine, and colon, respectively. The whole process is shown in Figure 1.

### 2.2. Proposed Organ Classification Method

#### 2.2.1. Organ Classification

Using the ResNet50 model [18] as a backbone network, the model is trained to classify three-class organs including the stomach, small bowel, and colon. This classification is a preprocessing step for detecting the start frame and the end frame of the small intestine. To adjust class imbalance among different organs, data augmentation or downsampling was conducted for each organ class so that the ratio of stomach, small bowel, and colon images is approximately 1:2:1. For cases from normal and abnormal patients, stomach images were augmented by two-fold using the horizontal and vertical flip. For cases from normal patients, small bowel and colon images were downsampled with the ratio of 2/3 and 1/3, respectively. For cases from abnormal patients, small bowel and colon images were downsampled with the ratio of 3/4 and 3/7, respectively. The number of the final training set images was 12,326,713, consisting of 3,144,548, 6,192,886, and 2,989,729 images for the stomach, small bowel, and colon, respectively. Our method was implemented in PyTorch and our models were trained on an NVIDIA RTX A5000 D6 24GB GPU. Adam Optimizer with a learning rate of 0.001 and cross entropy (CE) loss was used. Categorical classification (i.e., softmax activation), the standard categorical CE loss, is given by the equation:
(1)$$L_{ce}=-\frac{1}{N}\overset{N}{\underset{n=1}{\sum }}\overset{C}{\underset{c=1}{\sum }}y_{c}^{n}\times \log\left(h_{\theta }\left(x_{c},n\right)\right)$$ where *N* is the number of training examples, *C* is the number of classes, $y_{c}^{n}$ is the target label for training example n for class c, x is the input for training example c, and $h_{\theta }$ is the model with neural network weights θ.

The batch size is set to 128, and 224 × 224 resizing is applied. We report the test results extracted by the model that showed the best performance on the validation set.

#### 2.2.2. Transition Frame Detection

At the inference time, given a WCE video, our goal is to find the two transition frames for stomach to small bowel and small bowel to colon. First, the classification model (i.e., ResNet50) outputs probabilities of three classes (i.e., stomach, small bowel, and colon) for each frame. We only use the probability from “small bowel” to apply our temporal filtering method. The WCE video is fed to the pre-trained CNN as an input in chronological order to obtain the small bowel probability values, and then the temporal filtering is applied. The results of Savitzky–Golay filtering [19] and median filtering are added and halved, then values greater than 1 are mapped to 1, and values less than 0 are mapped to 0. If the maximum value is less than 1, values are divided by the maximum value. The filtering range is set to 1,001 frames for each filter. After filtering, the small bowel and the other two organs are separated by the threshold value of 0.87, which is obtained empirically. Then, it is determined that the minimum frame index being predicted as the small bowel is the beginning of the small bowel region, and the maximum frame index is determined as the end of the small bowel region. Finally, since the front and rear regions of the small bowel are stomach and colon regions, respectively, the three organs are classified through the proposed method. The whole process of the proposed method is shown in Figure 2. The code will be available on our GitHub page (https://github.com/MAILAB-yonsei, accessed on 22 June 2022).

#### 2.2.3. Relevance-CAM

To analyze where the trained 2D CNN model (i.e., ResNet50) focused on endoscopic images to classify organs, a relevance-weighted class activation map (Relevance-CAM) [20], which is an explainable model [20,21], was applied to ResNet50. The Relevance-CAM was extracted from the feature maps (i.e., the output of the convolutional and pooling layers) for the predicted class and then resized to the WCE image size (i.e., 320 × 320) and overlapped onto the original WCE image.

#### 2.2.4. Metrics

The performance of the proposed method is quantitatively analyzed in terms of accuracy, precision, recall, and f1 score.

Precision is the fraction of relevant instances among the retrieved instances, while recall is the fraction of relevant instances that were retrieved. Both precision and recall are therefore based on relevance, and f1-score is the harmonic mean of precision and recall.

## 3. Results

In Figure 3, Relevance-CAM results for representative organ images are shown. We were able to verify that the organ classifying process performed by the trained model was similar to those of endoscopists using the Relevance-CAM in which regions with clinical significance were indicated by the red color. We found that the model tends to classify organs through structural information such as wrinkles, the areas around dark areas captured along the track direction, submucosal vascular patterns, residual materials, and bubbles. In particular, bubble areas tend to be darkly weighted in small bowel images whereas they are brightly weighted in colon images because residual materials and bubbles frequently appear in the colon.

Table 1 compares the performances of Zou’s method [17], ResNet50, TeCNO [22], MS-TCN++ [23], and our proposed method. All metrics listed in Table 1 show the classification performances for the stomach, small bowel, and colon. While Zou’s method and ResNet50 are methods that utilize only single frame information, TeCNO and MS-TCN++ perform the classification using additional video information. Although the ResNet50 model is used for three-class classification before temporal filtering, the process changes to binary classification for small bowel detection (i.e., whether an image is from the small bowel or not) after the filtering. Nevertheless, since we know that the stomach and colon exist in front and rear of the small bowel, respectively, it is possible to determine the location of the small bowel using the three-class classification model.

As a classification model, ResNet50 shows better performance compared to Zou's method. In addition, when the temporal filter is applied, the accuracy and other classification performance metrics clearly become higher. ResNet50 with the temporal filter shows almost perfect classification performance in the other metrics. Even the proposed method shows higher performance than the method that utilizes video information.

Figure 4 shows the results of the proposed method for six cases randomly chosen from the test set. Through the entire process, the transition points were obtained, and the frame errors compared with the labeled marks were calculated. The transition time errors are shown in the bottom graph for each case. We found that the temporal filtering is very robust to misclassified images. As shown in all cases in Figure 4, after temporal filtering, the misclassified images (i.e., of which probability is close to 0 for small bowel or close to 1 for the other organs) are dramatically reduced. In particular, the second and third cases in Figure 4 show lots of misclassified frames in the colon region. Therefore, the filtered probabilities in the colon region are also relatively high (i.e., larger than 0.5). However, it was possible to differentiate the images to the small bowel or colon by proper thresholding. We set the threshold value to 0.87, which showed the best performance for the validation dataset. It can be interpreted that the temporal filter corrects the wrong classification using the probabilities that can be considered as features of length one estimated from adjacent frames. 

Time errors for each case can be calculated with the frame errors and fps (which is three for all test cases) for each case. Figure 5 shows the time errors calculated from the two transition points for stomach to small bowel and small bowel to colon. The time error for the transition between stomach and small bowel is averagely 38.81 s with a standard deviation of 25.8 s, and the time error for the transition between small bowel and colon is averagely 32.04 s with a standard deviation of 19.1 s. For the 40 test cases, there were only 9 cases where the transition time error was greater than 1 minute. Moreover, all transition time errors were less than 2 min. The results for each case are shown in Appendix A.

## 4. Discussion

The proposed small bowel detection method for WCE was developed based on the clinician’s marked frames for the two organ transition points, including stomach to small bowel and small bowel to colon boundary frames. The proposed method comprises a deep learning model (i.e., ResNet) followed by a temporal filter (i.e., the combination of a median filter and a Savitzky–Golay filter). For training and testing, a total of approximately 24,000,000 WCE images from 200 cases were used. Testing of the algorithm resulted in a high accuracy of 99.8% for the three-class organ classification and the average time error of 35.4 s for the organ transition frame prediction. Through this, we confirmed agreement between organ transition frames before and after small bowel predicted by the proposed method and those marked by WCE readers.

The threshold value of 0.87 for filtered probability values shows the best accuracy and robustness for the validation dataset. The reason why the value was set to higher than 0.5 is that, in specific cases, especially in the colon region, there are many images misclassified as small bowel, so the filtered probability values are higher than 0.5 for those colon images (e.g., Figure 4). In this case, when a threshold of 0.5 was chosen, the accuracy was considerably reduced because of the misclassified frames in the colon region. On the other hand, since the number of frames in the small bowel region that were misclassified by ResNet50 was relatively small, the filtered probability values were close to one as shown in Figure 4. Therefore, even when a threshold value of 0.87, which is close to one, was set, there was no case in which the small bowel images were misclassified in all validation or test sets.

As there is no standard algorithm for the classification of digestive organs, it can be difficult to judge the importance and performance of the developed algorithm. Therefore, it is critical to compare the algorithm with the clinically classified organs that are already in use by clinicians. We focused on how close the algorithm can predict the organ transition times compared to those marked by WCE readers. Furthermore, since there was no public WCE dataset for the corresponding organ classification task, data collection and annotation were also performed at our local institution (i.e., Dongguk University Ilsan Hospital). 

The use of the dataset obtained from only one vendor (i.e., Intromedic Co., Ltd., Seoul, Korea) is an experimental limitation of this study. Moreover, because the number of cases used in this study (i.e., 260 cases) is relatively small, more extensive studies are required to validate the clinical usefulness of the proposed method. Another limitation to consider is that WCE cases were acquired for a long time span (i.e., 20 years) and it is uncertain how this will affect the generalizability of the proposed AI model and results. In our future work, datasets from other vendors and multi-center data will be obtained so that multi-center studies or meta-analysis can be conducted to validate the generalizability and usefulness of the proposed method. Through this, the clinical validity would be more thoroughly confirmed.

Various algorithms, such as automated disease detection [6,24], a frame reduction system [25,26], cleansing score determination [27], and 3D reconstruction for the small bowel [28], have been developed to reduce the WCE reading time or increase convenience of reading. It is expected that these technical advances will greatly reduce the clinician's reading time and fully automate the clinician's diagnosis process, but there are several challenges, one of which is that the small bowel region must be extracted before those algorithms are applied. This is because most of the developed WCE disease detection algorithms are focused on the small bowel and are learned with only small bowel data, so it is difficult to apply those algorithms to full-length WCE videos that include frames from other organs. Therefore, only small bowel frames from an input full-length WCE video must be extracted first, so that the developed algorithms can be automatically applied. In conclusion, our small bowel detection method can be seen as a pre-processing step for the algorithms targeting small bowel. It is expected that it will be possible to fully automate the detection and analysis of small bowel diseases for WCE by combining the proposed method with the aforementioned automated algorithms.

Performance can also be improved by extending the proposed method to the temporal segmentation method that can utilize video information [23]. It is also expected to be used in cases such as surgical video phase recognition [22,29].

## 5. Conclusions

In this study, we propose a small bowel detection method for WCE using a CNN with the temporal filtering method. The time errors of organ transition were averagely 30 seconds and all errors were less than two minutes. Consequently, we demonstrate that the small bowel regions detected by the proposed method were highly correlated with clinical organ transitions marked by clinicians.

For many clinical cases, it is hard to specify where the transition between organs is because of various factors including very dark regions due to bleeding or melena, undetectable mucosa due to residual materials in the digestive organs, etc. Nevertheless, our method enables accurate small bowel localization in most WCE cases including abnormal cases such as bleeding, inflammatory, and vascular diseases.

## Figures and Tables

**Figure 1 diagnostics-12-01858-f001:**
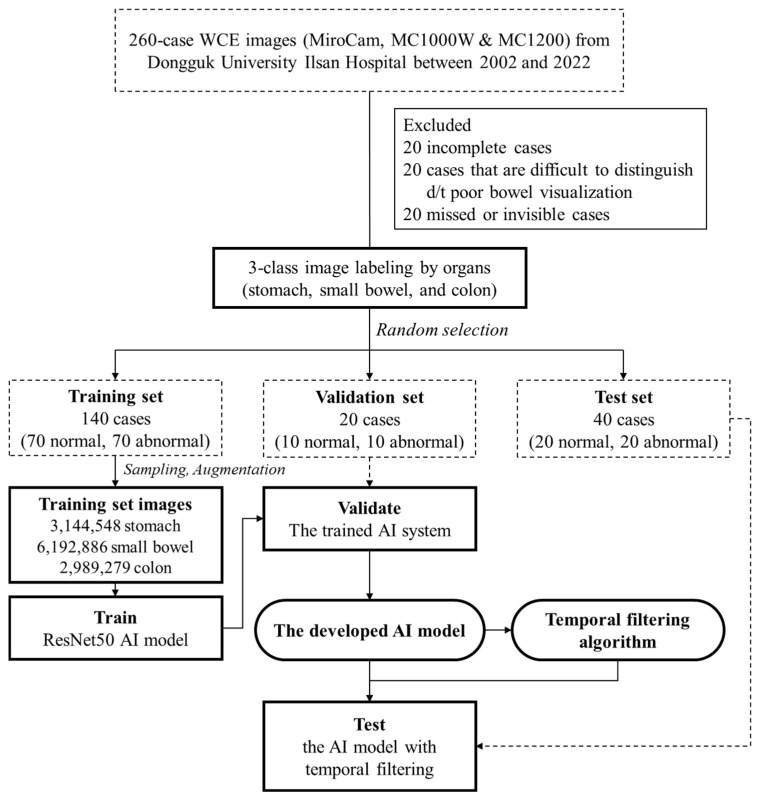
Flowchart of the study design.

**Figure 2 diagnostics-12-01858-f002:**
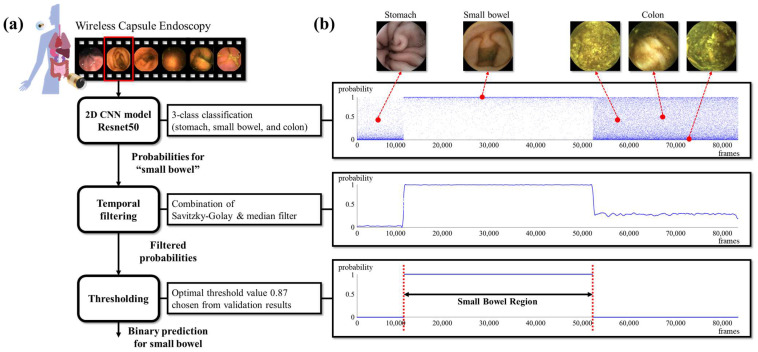
(**a**) The process of the proposed small bowel detection algorithm; (**b**) results from subprocesses with representative images from organs.

**Figure 3 diagnostics-12-01858-f003:**
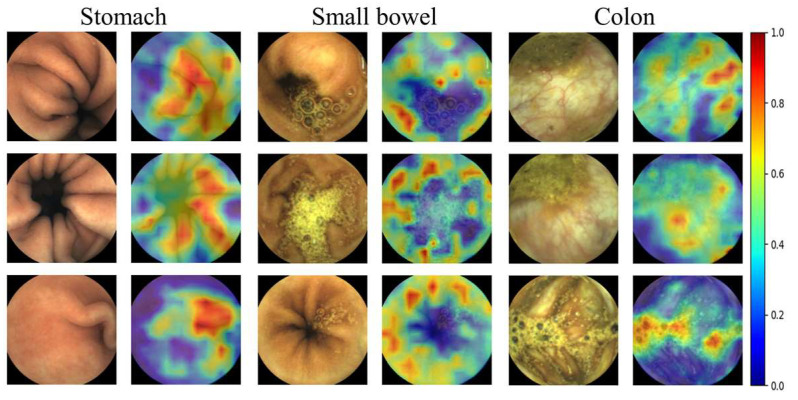
Relevance-CAM results from three organs. (WCE images taken by MiroCam MC1200 and processed by MiroView 4.0—http://www.intromedic.com/eng/main, accessed on 22 June 2022). The jet color maps show class activation maps in which the reddish and bluish colors refers to 1 and 0, respectively.

**Figure 4 diagnostics-12-01858-f004:**
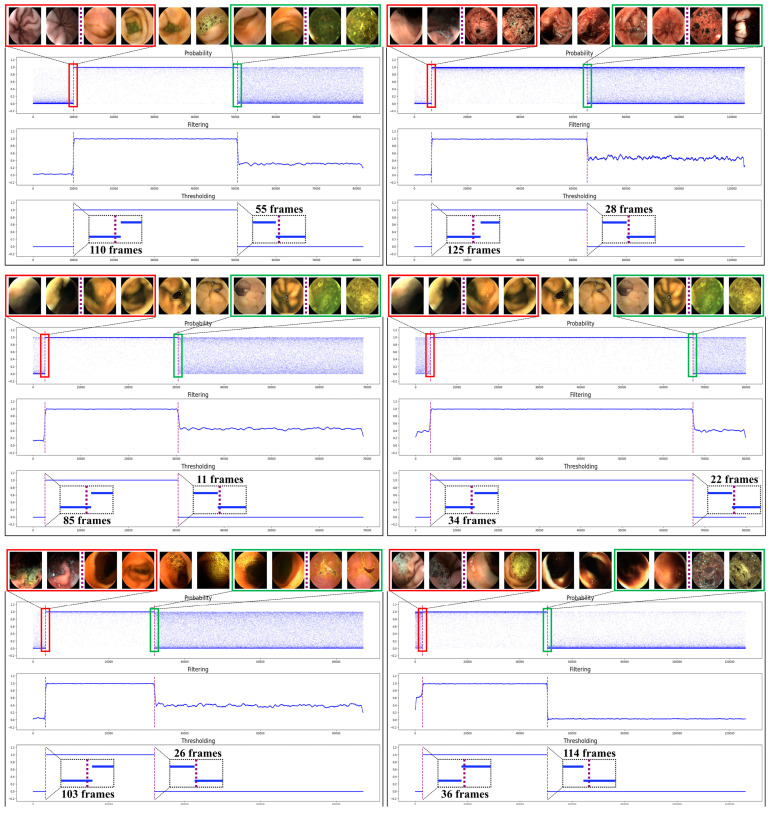
Qualitative results of the proposed small bowel detection method from six cases. Please refer to the Appendix A to see the results from all test cases.

**Figure 5 diagnostics-12-01858-f005:**
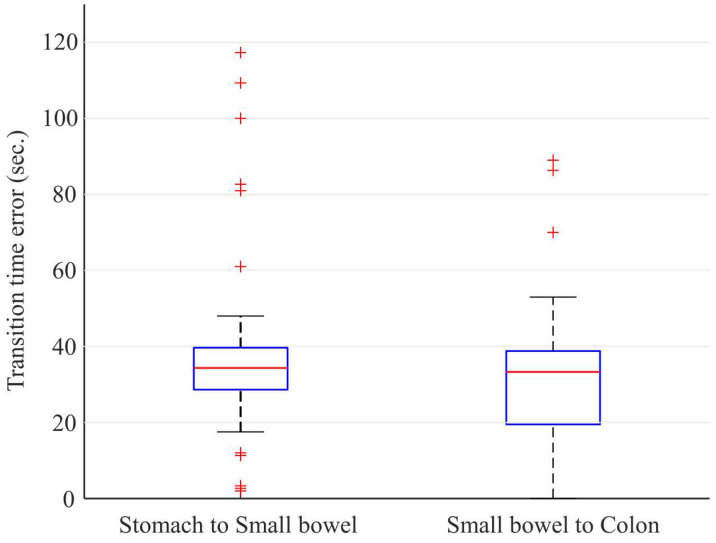
Time errors for the transition between stomach and small bowel and the transition between small bowel and colon. For all 40 test cases including 20 normal and 20 abnormal cases, all errors were less than 120 s.

**Table 1 diagnostics-12-01858-t001:** Quantitative results for organ classification. The proposed method (i.e., ResNet50 + temporal filter) yields the best performance.

Methods	Accuracy	Precision	Recall	F1 Score
Zou’s	0.751	0.689	0.768	0.712
ResNet50	0.880	0.876	0.872	0.872
TeCNO	0.900	0.920	0.873	0.892
MS-TCN++	0.937	0.941	0.896	0.869
**Proposed**	**0.998**	**0.998**	**0.998**	**0.998**

The highest values for each metric are bold-faced.

## Data Availability

Not available.

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
