# Peer review of "Small Bowel Detection for Wireless Capsule Endoscopy Using Convolutional Neural Networks with Temporal Filtering"

_diagnostics, 2022, doi:10.3390/diagnostics12081858_

Round 1
Reviewer 1 Report
This is an interesting study looking at the concept of automatically classifying the stomach, small bowel and colon in WCE images. This would have important clinical utility not only by reducing reading time but also as a pre-processing step before applying automated small bowel lesion detection algorithms.
The methodology is sound and includes a training set, validation and testing sets. The number of cases is rather small and it is a single centre study. The study captures a significantly long timespan (2002-2022) during which I would imagine that technological advances in capsule endoscopy occured and it is unclear whether this could affect the results. In this study only the Mirocam system was used, and it would be useful to assess the performance of the proposed method in other systems (such as Pillcam, which is more widely used).
Overall, I think the paper is well written; the topic is of interest and the study is good and well designed despite its limitations. I would suggest that the authors add a clear paragraph in their discussion with the limitations of the study for the benefit of the readers.
Author Response
We thank the reviewer for very considerate and constructive comments. We sincerely appreciate the time and effort spent for the review of our manuscript. We clearly annotated the items that were reflected in the manuscript in response to reviewer’s comments. All modifications were also clearly marked up using the “Track Changes” function in the revised manuscript. We added memos in the revised manuscript so that reviewer clearly check which comments are reflected.
We thank the reviewer and editor again for giving us a chance to refine our paper.
Sincerely,
Dosik Hwang, Ph.D. School of Electrical and Electronic Engineering Yonsei University Seoul 03722, Korea Phone number: +82) 2-2123-5771 E-mail: dosik.hwang@yonsei.ac.kr |
Yun Jeong Lim, Ph.D. Department of Internal Medicine Dongguk University College of Medicine Ilsan, Gyeonggi-do 10326, Korea
E-mail: drlimyj@gmail.com |
Reviewer 1’s comments:
This is an interesting study looking at the concept of automatically classifying the stomach, small bowel and colon in WCE images. This would have important clinical utility not only by reducing reading time but also as a pre-processing step before applying automated small bowel lesion detection algorithms.
The methodology is sound and includes a training set, validation and testing sets. The number of cases is rather small and it is a single centre study. The study captures a significantly long timespan (2002-2022) during which I would imagine that technological advances in capsule endoscopy occured and it is unclear whether this could affect the results. In this study only the Mirocam system was used, and it would be useful to assess the performance of the proposed method in other systems (such as Pillcam, which is more widely used).
Overall, I think the paper is well written; the topic is of interest and the study is good and well designed despite its limitations. I would suggest that the authors add a clear paragraph in their discussion with the limitations of the study for the benefit of the readers.
Point by Point responses to Reviewer1:
R1.1. The number of cases is rather small and it is a single centre study. The study captures a significantly long timespan (2002-2022) during which I would imagine that technological advances in capsule endoscopy occured and it is unclear whether this could affect the results. In this study only the Mirocam system was used, and it would be useful to assess the performance of the proposed method in other systems (such as Pillcam, which is more widely used). the topic is of interest and the study is good and well designed despite its limitations. I would suggest that the authors add a clear paragraph in their discussion with the limitations of the study for the benefit of the readers.
à As the reviewer's comment, we added a new paragraph about the limitations of this study in the revised manuscript as follows. (line#247)
“However, the use of the dataset obtained from only one vendor (i.e., Intromedic Co. Ltd., Korea) can be seen asis an experimental limitation of this study. AlsoMoreo-ver, because the number of cases used in this study (i.e., 260 cases) is ratherrelatively small, more extensive studies are required to validate the clinical usefulness of the proposed method. and only the MiroCam system was used. Another limitation to con-sider is that WCE cases were acquired for a long time span (i.e., 20 years)span and it is uncertain how this will affect the generalizability of the proposed AI model and results and can also be considered a limitation. In ourthe future work, datasets from other vendors and multi-center data will be obtained so that multi-center study or me-ta-analysis can be conducted to validate the generalizability and usefulness of the proposed method. Through this, the clinical validity wouldill be more thoroughly con-firmed.”
Reviewer 2 Report
In this paper, the authors used the ResNet-50 to classify three organs (stomach, small bowel, and colon), and then used a hybrid temporal filter to obtain the temporal probabilities for the “small bowel” class so that to distinguish the other two organs. The experiments demonstrated the proposed method has achieved promising results in identifying three organs. However, this paper has some major concerns before being published as the following:
1. In the introduction, the authors should summarize some latest research works using deep learning-based methods for organ classification with wireless capsule endoscopy. Compared with these research works, what are the advantages of the proposed method?
2. As we know, the ResNet-50 can directly divide gastrointestinal tracts into three organs: stomach, small bowel, and colon. Therefore, why further use the hybrid temporal filter for organ classification?
3. There are many popular CNN-based classification models including AlexNet, VGGNets, DenseNets, etc. Why do you choose the ResNet-50 to classify three organs? Compared with these models, what are its advantages in classifying three organs?
4. As the author said, “Since organs were filmed in order, if the small bowel can be accurately classified, the stomach and colon are also classified.” (lines 74-75). Since each examination is in the order of stomach, small bowel, and colon, I think an endoscopist can distinguish these organs quickly without errors. Why do the authors need to classify them?
5. The dataset collected by the authors consisted of 70 normal cases and 70 abnormal cases. Do the authors consider detecting organ lesions rather than just classifying organs? It is more important for endoscopists to determine whether there is any lesion in organs rather than only distinguishing organs. Therefore, does the proposed method still fail to shorten the reading time for endoscopists? So is it limited in the clinical application?
6. For “data augmentation or downsampling was conducted for each organ class so that the ratio of stomach, small bowel, and colon images is approximately 1:2:1”, please give more description in data augmentation or down-sampling.
Round 2
Reviewer 2 Report
Recommended to accept. Please add a loss function in the manuscript.
Author Response
We thank the reviewer for the positive decision.
We added the loss function (categorical cross entropy) in the revised manuscript.